# Assessment of Communication Abilities in Four Children with Early Bilateral CIs in Clinical and Home Environments with LENA System: A Case Report

**DOI:** 10.3390/children9050659

**Published:** 2022-05-04

**Authors:** Arianna Colombani, Amanda Saksida, Sara Pintonello, Federica De Caro, Eva Orzan

**Affiliations:** Institute for Maternal and Child Health-IRCCS “Burlo Garofolo”—Trieste, 34137 Trieste, Italy; sara.pintonello@burlo.trieste.it (S.P.); federica.decaro@burlo.trieste.it (F.D.C.); eva.orzan@burlo.trieste.it (E.O.)

**Keywords:** early cochlear implant, LENA, video analysis, language assessment, remote monitoring

## Abstract

Children’s language acquisition is underpinned by the quantity and quality of linguistic stimulation. Early diagnosis and cochlear implantation (CI), along with a family-centered intervention, are regarded as critical factors in providing appropriate language stimulation and thus supporting successful language outcomes in children with sensory neural hearing loss (SNHL). Considering the lack of tools to assess early language skills and open issues regarding the early predictors of CI outcomes, our goal was to evaluate the potential usability of the Language ENvironment Analysis (LENA) system as an early assessment and/or predictive tool. Clinical video recordings, LENA home recordings, and vocabulary scores were used to assess the progression of communication abilities of four children with CIs (6–35 m.o.). The data revealed a positive correlation between the estimated mean length of utterance (EMLU), vocal conversational turns (CT) in clinical video recordings, and receptive vocabulary, as well as the CT count in LENA being a significant predictor of productive vocabulary. These findings lead us to conclude that the LENA system has the potential to be used as an additional (tele-)measure in the early assessment of communication abilities of children with CI, as well as as a tool in the research of early predictors of CI outcomes.

## 1. Introduction

Profound hearing loss in children causes delays in the development of spoken language, with severe impairment of daily communication, literacy, and educational achievements along with long-term implications for social interaction, educational attainment, employment opportunities, and quality of life [1]. The adult’s responsiveness in the infant-caregiver interaction has been seen as one of the most influential variables related to the infant’s socio-emotional development [2]. Caregivers also play a key role in guiding attention and association, providing infants with the opportunity to gain semantic knowledge and increase vocabulary size [3]. Early diagnosis and early CI implantation, along with a family-centered approach to the intervention, are considered of utmost importance for the positive language outcomes in cochlear implantation.

### 1.1. Early Language Acquisition and Rehabilitation

Language acquisition in children is related to the provision of abundant linguistic stimulation within a context of communicative interaction [4]. Children with hearing loss are considerably more vulnerable to this variability as they begin with a disadvantage, experiencing inadequate stimulation, access, and exposure to language. For these reasons, early diagnosis followed by prompt intervention [5] and family-centered interventions that encourage parents to expose their children to significantly more child-directed speech is recommended to minimize these risks. While the assessment of language skills is conducted regularly in order to assure timely follow-up, an ecological evaluation of the home language environment and child’s language skills outside the laboratory settings represents a challenge.

Whereas the age of hearing aid fitting is not associated with strong outcomes, the age at CIs is one of the significant predictors of positive language outcomes in children with hearing loss [6]. Children who receive implantation before 2 years of age have significantly better language outcomes compared to older children [7]. Hearing screening by 1 month of age, diagnosis of hearing loss by 3 months of age, and enrollment in early intervention by 6 months of age are therefore the universal guidelines recommended by international committees [8].

This progressive reduction of the age at implantation has led to a novel demand for different assessment methods, which should be objective, ecological, and suitable to evaluate very young children with cochlear implants at a very early stage of their rehabilitation path. Namely, during a clinical speech-language evaluation session, it might be challenging to obtain information about the authentic language abilities of a child and the natural communication environment in his/her everyday life. Multiple factors such as child fatigue and lack of interest, the presence of new people, unfamiliar surroundings along with parents’ concern about their own “performance”, can interfere with the process. One such possible additional measure in the early assessment of communication abilities of children with hearing loss is the automatic analysis of home day-long recordings, such as the one provided by the Language ENvironment Analysis (LENA) system, which offers an overall estimate of a child’s home language and auditory environment and the rough estimate of a child’s communication skills [9,10,11].

### 1.2. CI Outcomes Variability

While there is no longer any disagreement about the general efficacy of cochlear implants and the benefits of early implantation, there is still great variability in the effectiveness of CIs. One long-standing challenging issue in this field is understanding and explaining the underlying mechanisms that lead to the major variability in the speech and language outcomes following implantation [12].

Both pre-implant and post-implant factors have been explored in this regard. Tait, Lutman, and Robinson [13] used video-analysis to correlate the percentage of turn-taking and autonomy before implant with scores at standardized tests of speech production and perception tests at 3 years after implantation, showing that pre-implant autonomy (either vocal or gestural turn-taking) positively correlates with speech identification and production. Castellano et al. [14] carried out MacArthur–Bates Communicative Development Inventories (MB-CDI) to investigate the predictive value of early expressive vocabulary for long-term language and neurocognitive outcomes of CIs. The study shows that MB-CDI can be a reliable early index of long-term neurocognitive skills, including language, verbal working memory, fluency speed, attention, sustained sequential processing, and basic reading and writing skills. The strong relationship between vocabulary growth and language skills has long been recognized. As shown by Rowe, Raudenbush, and Goldin-Meadow [15], children’s vocabulary at school entry is predicted by the pace of early vocabulary growth which is, in turn, influenced by parent education and family income (SES), parent vocabulary input, and the child’s gesture vocabulary at 14–18 months, also predicting language skills either as production [16] and vocabulary comprehension [17].

The number of words or morphemes in each of a child’s spontaneous utterances, also known as the measure of the mean length utterance (MLU), is another well-established indicator for describing children’s language abilities [18]. In clinical application, MLU has been identified as an important predictor of language outcomes for children with autism [19] and as a general predictor of language skills in children with SLI [20]. MLU may therefore prove to be a good evaluation measure also in children with CIs. A reliable estimate of MLU may be difficult and time-consuming to obtain through standardized (clinical) tests of language abilities, as it is traditionally computed by taking 100 utterances from a child and dividing the number of morphemes by the number of utterances. Estimated MLU is one of the outcome measures of the automatic analysis of the LENA system’s day-long recordings.

Additionally, the quantity and quality of child-directed speech [21,22,23,24], lexico-syntactic diversity [25], decontextualized language (narratives, explanations, engaged and pretend play; see [26]), along with parents’ sensitivity and responsiveness to child’s requests [27,28,29,30] are all crucial factors for successful language outcomes. Some of these characteristics of communication can be successfully measured with the automatic analysis of day-long recordings. For example, parents’ sensitivity and responsiveness to a child’s requests can be measured in terms of conversational turns, and quantity of child-directed speech can be estimated by the amount of adult speech recorded in close proximity to the child.

### 1.3. Objectives

In the present study, we sought to address the above-discussed issues related to the dearth of assessment methods and predictors of language outcomes in young children with CIs by exploring the relation between speech and language measures and novel methodologies, such as video analysis and LENA day-long recording systems. In order to control for other factors that are known to contribute to language outcomes, such as neurocognitive development and SES, the longitudinal data from a small but highly homogeneous sample of four children with early bilateral CI implantation were examined. We measured the extent to which data obtained through LENA recordings correlated with other clinical tools for the early assessment of language development and explored whether these data could be used as a reliable additional tool to assess children’s early communicative skills and as a predictor of their language skills.

## 2. Materials and Methods

### 2.1. Participants

Four children, two boys and two girls, were recruited between 6 and 15 months of age, among the cases followed in the Department of ORL and Audiology at the Institute for Maternal and Child Health IRCCS “Burlo Garofolo” in Trieste, Italy. The four children were selected in order to create a homogeneous study sample to better evaluate the assessment methods without any influence of other variables.

All four children (C003, C004, C005, and C006) presented profound congenital sensorineural hearing loss (SNHL), diagnosed through the neonatal hearing screening program held at the Institute. None of the children presented cognitive delay, comorbidities, or additional disabilities. For all cases, deafness had a genetic but non-syndromic etiology.

All four participants were fitted with bilateral hearing aids, followed by early bilateral cochlear implantation. All of them received the first implant (to the right ear) at 11 months. Children C004 and C005 received the second implant at 12 months, while children C003 and C006 at 15 months, with a continued bimodal stimulation throughout the process. The implant type was a Cochlear CP 1000 (C004 and C005) or a Medel with a Sonnet processor (C003 and C006). The children came from hearing parents, without a family history of hearing loss. All families presented an average medium-high SES level with a high maternal educational level (university degree). None of the children had siblings. All children underwent regular fitting and evaluation procedures and have been regular all-day users of the implants since the implantation.

Their neurocognitive abilities were measured at one year after the first implant. All four children had an average or above average non-verbal intelligence quotient (IQ) (range: 100–132), while verbal IQ was below average (C003: 77) for one child and average for the remaining three children (range: 94–114).

Prior to the recordings, parents gave their written consent to participate in the day-long home recording study. They were also informed about the usage of clinical data for research purposes and gave their written consent to participate before the assessment (see Appendix A). The study was conducted in accordance with the 1964 WMA Helsinki declaration and its later amendments, under the framework of the research project 17/17 approved by the institutional ethical review board, nominated by the Italian Ministry of Health (Ufficio per la Ricerca Clinica IRCCS Burlo Garofolo). 

### 2.2. Language/Vocabulary Assessment Tool

In assessing the auditory and language abilities of very young children affected by SNHL, few standardized methods are available. Three assessment tools were used to evaluate receptive and productive vocabulary in the study sample. 

For the assessments up to 18 months of age, the Italian version of the MacArthur–Bates Communicative Development Inventories (MB-CDIs) [31], called PVB_Primo Vocabolario del Bambino test [32], was used. The questionnaire is used for the assessment of communicative-linguistic development, from non-verbal components to grammatical development, accurate and valid in children with typical development as well as in atypical populations, including deaf children with CIs. Additionally, a direct language observation was conducted using the standardized test PinG Parole in gioco [33] (18–36 m.o.). The test was used to evaluate the comprehension and lexical production of nouns along with the understanding and production of predicates and consists of a series of color images with names or predicates that must be comprehended by pointing and naming. The Test Fono Lessicale (TFL) [34] (>36 m.o.) was administered to further assess the receptive and productive vocabulary. The test is comprised of 45 tables, each containing four images: the target, a phonological distractor, a semantic distractor, and a non-related distractor. The examiner pronounces the target word and asks the subject to point to the image depicted by the word uttered. Using the same pictures, the lexical production test is used to evaluate the ability to find the proper lexical label both directly and after receiving coded semantic and/or phonological guidance. In all three assessment tools, the results were compared to the standardized scores, and percentile ranks were measured for each test administration.

### 2.3. Tait Video Analysis

The video analysis method provides the clinicians with a 15 min long recording of a dedicated play between a child and a caregiver in a semi-ecological (laboratory) setting. From the recording, preverbal communications skills can be analyzed (including appropriate eye contact, conversational-style turn-taking, auditory autonomy, and awareness of the sound of speech) along with the caregiver’s communicative style. For clinical practice, the first 20 communicative turns are analyzed, and the percentage of auditory autonomous non-looking vocal turns is computed.

The method of Tait video analysis is a technique for assessing preverbal communication behaviors in very young children with hearing impairment and has been found to be strongly related to speech discrimination and intelligibility outcomes post-implantation [35,36]. In children with acoustic hearing aids and cochlear implants, the method has been shown to be a reliable tool for assessing [37] and monitoring [36] pre-verbal language skills, and to correlate well with other measurements [38]. As a result, the method has been implemented into everyday clinical practice in various countries.

### 2.4. Language ENvironment Analysis (LENA)

Language ENvironment Analysis (LENA) is a proprietary automatic speech processor (ASP) device for collecting and analyzing day-long audio of children and families in their natural home environments. First developed in the mid-2000s by the LENA Research Foundation (Boulder, Colorado, United States), it gained prominence among clinicians and researchers interested in children’s language development.

The LENA system consists of a digital language processor, endowed with automatic processing software. The child wears a small recording device (Digital Language Processor, DLP), which records day-long audio data of the child in his/her natural context with his/her communication partners, collecting spontaneous, unrehearsed, and representative samples of the child’s typical daily language environment, providing researchers with ecologically valid samples of data with consistent formats across laboratories. The resulting audio file is subsequently processed by the software, which automatically analyses the recordings and assigns a sound class to each audio sample (speech, silence, TV). Audio segments labeled as overlapping speech, silence, noise, are not taken into consideration while speech segments are further vetted, yielding the following variables: a count of adult words (AWC), key child vocalizations (CVC), conversational turns (CT), and estimated mean length of utterance (EMLU). To establish LENA reliability, a number of studies have compared the labels that LENA assigns to audio segments to those assigned by human transcribers, indicating reasonable levels of agreement for AWC and CVC [39] and for CT and EMLU [40].

To establish LENA reliability, a number of studies have compared the labels that LENA assigns to audio segments to those assigned by human transcribers, indicating reasonable levels of agreement [10,39,41,42,43,44,45]. Although no studies on LENA reliability for the Italian language were found, a good correlation between LENA adult word count (AWC) and manual scores has been shown in languages close to Italian, such as Spanish (as spoken in the USA, [46]) and French [47]. Because of this, we can assume certain reliability for the Italian language as well.

The LENA System has been used extensively to investigate various aspects of parent–child talk and interaction, including the positive relationship between language development and the number of conversational turns [45] and language environment [46]; and the positive relationship between the number of conversational turns and the number of vocalizations [48]. It was also used to gain insights into the auditory environment and language development of clinical populations, including children with hearing loss [49].

### 2.5. Procedure

Families of the four children were asked to record the child’s auditory environment one whole day every three months using the LENA recording device placed on the child’s chest. All recordings were collected in the children’s home environment and included all vocalizations produced by the key child wearing the device and all external sounds or vocal activities within a range from 1.5 to 2 m. A total of 11 recording sessions were collected for the child C004, 10 for C005, 8 sessions for the child C003, and 7 for C006, for an average length of 10.4 to 12 h per child. In order to better compare the language production level reached during video analysis in the clinical setting with the naturalistic production in the home environment, the most active hour of the day was selected and processed using the LENA software. The following statistics related to the child’s auditory environment were selected: number of child vocalizations (Child.voc.LENA), the estimated mean length of the utterance (EMLU.LENA), the number of conversational turns (Turns.LENA), and the number of words produced by the adults (AWC.LENA).

The data acquired from these home audio recordings were then analyzed together with the data from (clinical) video analyses, from which the percentage of vocal communicative turns (independent from the visual information) was taken. Video analyses were made during each follow-up exam, during which auditory, language, and communicative skills were also measured as a part of comprehensive speech therapy exam. The follow-up exams were scheduled at roughly the same time as the recordings with LENA. In total, 5 reports from follow-up exams were analyzed from participant C003, 8 from C004 and C005, and 6 from C006. Receptive and productive vocabulary variables were obtained from these medical reports, as well as the percentage of vocal turns from the videoanalyses (Turns.videoanalysis). Additionally, the recordings obtained for the video analyses were fully transcribed in order to obtain information about the number of adult speech production (AWC.videoanalysis), child vocalizations (Child.voc.videoanalysis), and the estimated mean length of utterance for each video (EMLU.videoanalysis). The raw dataset with all the included variables is available in the Appendix A.

### 2.6. Data Analysis

Overall differences between children were assessed with a series of one-way analyses of variance per each of the output variables. Non-parametrical Kruskal–Wallis tests were used for the variables that were not normally distributed. *p*-values were Bonferroni corrected for multiple comparisons. For each child, a multiple correlation analysis was performed between variables obtained by LENA recordings (AWC.LENA, Turns.LENA, EMLU.LENA, Child.voc.LENA), variables obtained from video analysis (Turns.videoanalysis) and the transcriptions (EMLU.videoanalysis, Child.Voc.videoanalysis, AWC.videoanalysis), and the variables related to vocabulary growth (Receptive and productive vocabulary).

Finally, two exploratory linear mixed effect models were created to assess possible predictive value of the above variables for Productive and Receptive vocabulary in all 4 children (consisting of 7–11 time points per child; see Appendix A). In both, fixed factors were Turns.videoanalysis, EMLU.LENA, and EMLU.videoanalysis, the variables that correlated with vocabulary measures in the four participants. The last fixed factor (Child.age) and the structure of the random factor (Child.age|Child.ID) accounted for the longitudinal sampling. Statistical analysis was performed in R software [50].

## 3. Results

One-way analyses of variance between the four participants were computed for all output variables. No significant between-subject differences were found for any of the variables, except for the vocalizations estimates by LENA (Child.voc.LENA), where child C004 produced significantly less vocalizations compared to C005 (F(3) = 5.482, *p* = 0.026 (Tukey HSD post-hoc test, Bonferroni corrected for multiple comparisons)).

Multiple correlation analysis between variables from LENA recordings, video analysis, and vocabulary growth showed a strong positive correlation between EMLU.LENA and EMLU.videoanalysis (for all children R > 0.97, *p* < 0.005). Furthermore, in all four cases EMLU.LENA was positively correlated also with communicative turns as measured with video analysis (Turns.videoanalysis) and Receptive vocabulary, while results in Productive vocabulary were more variable. These correlations, along with significance values, are represented in Figure 1. The correlations with AWC and communicative turns as measured with LENA were much less consistent across the four cases and did not reach significance. Correlation plots with correlation coefficients and indicated significance values for each child separately are presented in Appendix A.

To assess a possible predictive value of measures obtained by LENA and clinical video analysis, we created simple linear regressions with the variables that showed some correlation with clinical vocabulary measures in the previous analysis. Separate models for each child did not converge. We therefore created two exploratory linear mixed effects models with Productive and Receptive vocabulary in all four participants as dependent variables, respectively. In both, fixed factors were Turns.videoanalysis, EMLU.LENA, and EMLU.videoanalysis, with Child.age and the random factor (Child.age|Child.ID) accounting for the longitudinal sampling. For Productive vocabulary, beyond Child.age (Sumsq = 161.43, F(1,3.92) = 67.77, *p* = 0.001), EMLU.LENA (Sumsq = 209.76, F(1,0.96) = 88.06, *p* = 0.07) and Turns.videoanalysis (Sumsq = 352.90, F(1,0.97) = 148.15, *p* = 0.06) were marginally significant predictors. However, the variability among participants was larger than the magnitude of any of the fixed effects in the model, which indicates that the model was weak at predicting the results despite the relatively normal distribution and low autocorrelation of conditional residuals (see Appendix A for details). Conversely, Receptive vocabulary was significantly predicted by EMLU.videoanalysis (Sumsq = 441.89, F(1,4.11) = 10.20, *p* = 0.03), in addition to the effect of Child.age (Sumsq = 822.08, F(1,6.23) = 18.98, *p* = 0.004). The variability among participants was again larger than the magnitude of any of the fixed effects in the model, indicating weak fit of the model. Comparison with simpler model was not possible because they failed to converge. 

## 4. Discussion

The purpose of the present study was to examine the reliability of the LENA system as a possible assessment tool for early language abilities in very young deaf children using cochlear implants. Accordingly, we questioned the overall feasibility of using the LENA system as an assessment tool in the early intervention programs.

Through LENA, we collected and analyzed audio recordings from children’s home environments. The number of child vocalizations, the estimated mean length of the utterance, the number of conversational turns, and the number of words produced by the adults were taken as the out variables from LENA. We compared them to clinical measures obtained during regular follow-up exams: proportion of conversational turns from the video analysis and receptive and productive vocabulary measures obtained through parental questionnaires and standardized tests. We analyzed data from a small but highly homogeneous sample of 4 early-implanted children.

The analysis revealed three significant results: (1) a positive correlation between the EMLU found in LENA and the one from the video analysis; (2) a positive correlation between EMLU.LENA, the receptive vocabulary, and vocal conversational turns in three out of four participants; (3) the conversational turn count in video analysis as a marginally significant predictor for Productive vocabulary; (4) the EMLU estimated from video analysis as a significant predictor for Receptive vocabulary. Our findings are consistent with those of Tait, Lurman, and Robinson [13], who found a positive correlation between the number of turn-takings and later language outcomes, as well as with previous literature on the positive correlation between EMLU and language abilities [18]. 

The last two results are, however, to be taken with extreme caution, given that they result from mixed effect models built with only four participants and relatively small number of data points per participant (sometimes only four). Such a small sample size can severely skew the results. Therefore, the possible value of clinical video analyses and day-long recordings to predict language outcomes needs to be further explored in a larger sample size. Nonetheless, the present data represent a meaningful starting point for further research, also because, to our knowledge, only a few studies have been done to date on the possible implementation of automated analyses of daily recordings in clinical assessment, monitoring, and rehabilitation in young children with cochlear implants [11]. 

Somewhat surprisingly, no consistent correlations were found between the quantity of adult or child speech and language outcomes as has been found in previous work [51]. Again, all results are strongly affected by the small sample size and the small number of observations that were correlated (sometimes only four). In such a small sample, outliers have a significant impact on outcome. It remains for future research to explore this issue in a larger and more diverse group of children. Moreover, it would be of great interest to further investigate the longitudinal predictive value of measures obtained by day-long home recordings and explore possible relations between these measures and language outcomes later in childhood.

The present work provides some evidence that the estimated mean length of the utterance (EMLU) extracted from the analysis of LENA recordings can serve as a predictor for children’s receptive vocabulary. Regardless of the above-mentioned limitations of this four-case study, we predict that the LENA system could serve as an automatic counterpart to clinical video analysis, providing a different insight on child abilities by evaluating their in-home linguistic environment. Nevertheless, being based solely on the auditory inputs, it should be noted that LENA cannot provide an exhaustive image of the communicative environment, missing all the non-verbal behaviors such as gestural communicative turn-taking, eye contact, and gestural autonomy, which could be instead detected by a tool such as video analysis. Therefore, the synergic use of the LENA system with other methodologies could provide clinicians with a better overview of the infant’s communicative environment in its entirety. Identifying the best methods for conducting ecological and objective assessments of the impact of cochlear implantation on language outcomes would assist healthcare practitioners to optimize services for this population. Remote monitoring could provide a noticeable advantage in assessing children in their home environment, helping families to overcome the logistic challenges by reducing time and travel burdens [52] and, in turn, helping the service providers to manage the constantly increasing workload demand. Least but not last, language feedback on linguistic and communicative behaviors should also be offered to parents, as a part of any intervention. Home recordings could become a useful tool to gain insights into children’s auditory environment and provide quantitative feedback to caregivers of young children with hearing loss [53,54].

## Figures and Tables

**Figure 1 children-09-00659-f001:**
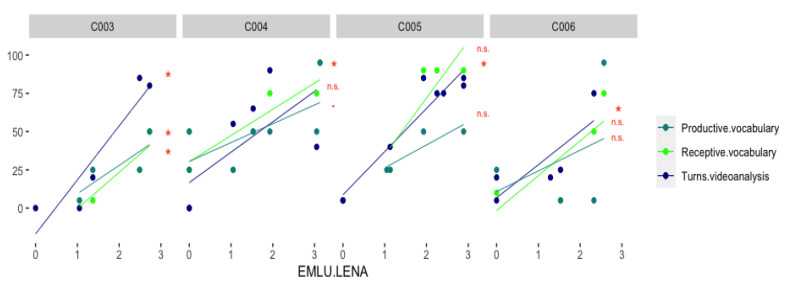
Percentile scores for Productive and Receptive vocabulary, and the percentage of vocal conversational turns estimated from video analysis as measured at different EMLU. The lines represent the linear regression estimates between each variable and the EMLU from LENA.

## Data Availability

The raw dataset presented in this study is available in Supplementary Material S1.

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
