# Peer review of "Assessment of Communication Abilities in Four Children with Early Bilateral CIs in Clinical and Home Environments with LENA System: A Case Report"

_children, 2022, doi:10.3390/children9050659_

Round 1

Reviewer 1 Report

Some general and major comments include:

  1. All acronyms should be defined on their first occurrence. For instance, CI (line 11), SNHL (line 13), MB-CDI (line 83), IQ (line 143) and CVC (or CV? line 212).
  2. Which software was used to perform the statistical analyses? Please cite it, e.g., in Section 2.6.
  3. Why don’t you perform a one-way MANOVA (multivariate analysis of variance) instead of “a series of one-way analyses of variance per each of the output variables” (lines 258-259)?
  4. Moreover, wouldn’t nonparametric tests be preferable for this application because of the small sample size?

Some suggested minor changes include:

Line 46. “... is conducted ...” instead of “... are conducted ...”

Line 61. “... in his/her ...” instead “... in her ...”

Line 76. “... to correlate ...” instead of “... to correlated ...”

Line 79. “... positively correlates ...” instead of “... positive correlates ...”

Line 82. “... long-term ...” instead of “... long term ...”

Line 87. “... [15], children’s ...” instead of “... [15] children’s ...”

Lines 101-103. Please revise the whole sentence.

Line 106. Please remove the period right after the word “see”.

Line 117. “... early assessment ...” instead of “assessment early”

Line 187. Please add a simple space right after the word “post-implantation”.

Line 200. “... his/her ...” instead of “... his ...”

Line 201. “... his/her ...” instead of “... his ...”

Line 215. “... [10],[39],[41]-[45] ...” instead of “... [10],[39 ],[41 ]-[45] ...”

Line 226. Please remove the extra word “a”.

Line 248. Please insert a comma right after “In total”.

Line 270. “One-way ...” instead of “One way ...”

Lines 273-274. Are these results (test statistic value and p-value) related to the ANOVA F-test or to a multiple comparison test (e.g., Fisher’s LSD test)? Please clarify this.

Lines 300-302. Please revise this sentence, once the normal distribution of residuals does not seem to be valid for the model with Productive vocabulary as the response variable (see Figure S3-2).

Line 384. “... of the ...” instead of “... of th ...”

Line 385. What does the symbol “?” mean in the multiple correlation plot for child C005?

Figures S3-1, S3-2, S3-3 and S3-4. Are these the marginal or conditional residuals?

Reviewer 2 Report

Please see file attached

Round 2

Reviewer 1 Report

The authors have satisfactorily addressed all my previous comments and now the paper can be accepted for publication.

Author Response

We thank the reviewer for the final assessment of the manuscript.